# Exploring Migration Hold Factors in Climate Change Hazard-Prone Area Using Grounded Theory Study: Evidence from Coastal Semarang, Indonesia

Choirul Amin [1,*], Sukamdi Sukamdi [2] and Rijanta Rijanta [2]

1   Faculty of Geography, Universitas Muhammadiyah Surakarta, Surakarta 57169, Indonesia
2   Faculty of Geography, Universitas Gadjah Mada, Yogyakarta 55281, Indonesia; sukamdi@ugm.ac.id (S.S.); rijanta@ugm.ac.id (R.R.)
*   Correspondence: ca122@ums.ac.id

**Abstract:** Though those who stay put in climate change hazard-prone areas are an intriguing subject of research, only a small number of empirical works specifically targeted these populations. Hence, the drivers of immobility in disaster-prone areas remain understudied and inadequately theorized. In response to these gaps, this contribution locates environmental immobility. The study aims to construct a theoretical model and examine the model through the evidence from the fishing community on the coast of Semarang, one of the areas most severely affected by tidal inundation in Semarang, namely Kampong Tambak Lorok. Using the study of in-depth substantial interviews from 24 participants, we use the grounded theory method to construct a theoretical model. The findings show that the grounded theory's coding process generated 18 initial concepts, eight main categories, and four core categories. It explores some of the reasons why populations continue to stay, even in the face of environmental degradation. There were two following conclusions: (1) Populations who stay put in disaster-prone areas are held by place attachment, family ties, social ties, and occupational ties. (2) Migration hold factors generate immobility by resisting the forces of migration push factor. The study meaningfully incorporates the migration hold factors as one of the drivers of immobility and enhances the field of environmental immobility theory, migration theory, and environmental migration research. Besides, some policy suggestions are provided as a result of the research findings. For future study, this research also offers a reference for exploring theoretical models of migration hold factors in other regions and countries with different environmental degradation settings.

**Keywords:** immobility; migration hold factor; grounded theory; climate change; tidal inundation; coastal Semarang

## 1. Introduction

There is growing attention to population immobility associated with environmental degradation regarding the possible relation between global environmental changes, extreme environmental occurrence, and human migration [1]. When environmental problems occur, the population does not necessarily migrate because they have three choices: stay in the disaster area and do nothing, stay in the disaster area and carry out disaster mitigation, or leaving the affected area (migrating) [2]. Sometimes people are forced to migrate, but most are not. Therefore, there is still much debate about how environmental problems will ultimately affect human mobility [3]. However, there is a link between environmental degradation and migration [4,5].

The integration of environmental migration theories into migration research has generally been inadequate [3]. Migration research has been sluggish in recognizing the significance of immobility for initiating, developing, and maintaining migratory projects. Migration studies tend to focus on migrants, i.e., people who migrate. Hence, migration studies pay less attention to people who do not migrate. There is only a little study of the

reasons why people do not migrate. Therefore, it is not surprising that existing migration theories generally ignore immobility and cannot explain the absence of migration [6–8]. Likewise, the research on migration and environmental change has focused more on population movement than lack of mobility [3].

The early work on population migration responding to environmental degradation focuses on migration as an adaptation strategy to environmental change. The research rightfully expands on how people try to deal with environmental change use mobility as an adaptive livelihood strategy [5,9,10]. The use of migration as an adaptation strategy to environmental change does not apply to everyone. Factors such as resources, information, social, and personal factors all influence the outcome. It is often the most severely affected and least able to migrate who are the most vulnerable [11]. While people who continue to stay in places impacted by environmental change are a significant conceptual concern, empirical work has been confined primarily to those communities. A limited number of researchers have commenced establishing a hidden feature within this research field: immobility [1,3,12].

In the literature, there is also an apparent existing movement towards environmental immobility. However, immobility drivers in a disaster-prone area remained understudied and inadequately theorized [8]. Immobility must be approached as a process with its own determinants to incorporate immobility into migration research meaningfully [7,8,13]. As an integral aspect of the continuum of mobility, the complexity of immobility and its drivers in crisis demands rigorous investigation. Concerning these discrepancies, this research describes disaster immobility in mobility studies. It demonstrates the significance of migration hold factors for understanding disaster immobility.

Considering the push and pull factors, migration is a rational response to the condition of the place of origin [14]. However, in some cases related to environmental degradation, some individuals do not migrate even though there is an adequate push factor in the form of environmental degradation. It raises an essential question in studying migration and environmental degradation: what is holding them back from migrating?

In delivering multi-causal understandings of environmental migration, the following study joins the vast majority of current studies and attempts to clarify the migration hold factor in areas experiencing serious environmental degradation. This study intends to fit our previous research on environmental migration rather than to militate. It plays a role in the body of literature by integrating the drivers that have a role to disaster immobility through a qualitative study, without diminishing other factors' role in 'holding people in place'.

This study highlights the importance of incorporating disaster immobility into migration studies to advance the research agenda on disaster immobility and establish theoretical and methodological advancement on how to do so. Therefore, we aim to promote a theoretical model to open up the black box of disaster immobility. So, what are the factors holding population mobility in a disaster-prone area? How does the hold factor model of disaster immobility work? The scientific challenges that this paper seeks to answer are these two significant concerns. We take the fishing community on Semarang coast as the research object and proceed to analyze in-depth interview materials from 24 informants. The theoretical model is then developed by grounded theory, putting down the groundwork for further study.

The paper has five sections. This introduction is in Section 1. As follows, the other four sections are: The literature review in Section 2. Section 3 deals with the materials and methodological aspects, the study location, the research methods, and the data sources. Section 4 is the analysis of the result, addressing: open coding, axial coding, selective coding, the test of theoretical saturation, and description of the model. The conclusion and implication are in Section 5. The core findings, policy implications, study limitations, and potential odds of this paper are set in motion by emphasizing some of the benefits of open up the black box of disaster immobility for migration studies.

## 2. Literature Review

Immobility and migration are in the same field. Some researchers have placed the theoretical framework for immobility research in the field of migration. The current study on immobility in a disaster-prone area is mainly born out of the impact of environmental change on migration. Moreover, disaster immobility research is not systematic and profound and cannot adequately explain the driving factors of disaster immobility. Focusing on immobility in disaster-prone area, this research mainly reviews the literature from the term of immobility, immobility as research topic, the immobility in environmental change, and the research on disaster immobility.

### 2.1. Immobility as a Research Topic

Researchers have used various terms to describe immobility ranging from non-migrants, staying put, involuntary immobility, and stayers [13,15,16]. Although immobility has acquired status as a proper research study, its depiction as a default circumstance still prevails in some migration literature. Stayers have also been labeled as 'left behind' [17].

Immobility is a growing area of research interest, even though it does not spread as many ideas as mobility [18,19]. Two significant study strands have included immobility: transnational family literature [20] and migrant networks literature [21]. The transnational family and the migration network have an analytical framework that includes migrants as well as non-migrants. However, there is still potential for complementary viewpoints on the frameworks that differentiate between those capable of realizing migration aspirations and those who do not migrate voluntarily or unintentionally [16].

Traditionally, migration theories address both voluntary and involuntary mobility. However, the age of migration cited by [22] has proved simultaneously to be the age of involuntary immobility. This analytical distinction created the potential to account for the broader fact that earlier theoretical models were unable to clarify: the involuntary immobility of individuals facing immobilization processes in the so-called age of migration that, despite their ability to do so, prevented them from migrating [7].

Although immobility is often invisible, it is inextricable and connected to our understanding of human mobility. Therefore, it is possible to see migration and non-migration as two sides of the same coin [23]. Mobility and immobility are always interrelated and interdependent. Both must always be interpreted together, not as opposites, but as complex constellations with multiple scales and concurrent practices [24]. Hence, exploring mobility means also zooms into the motives for immobility.

Immobility, similar to migration, often becomes constituted life strategies that engage changing articulations of mobility–immobility [7]. The key argument is that these components alone are inadequate to explain trends in real-world migration. It is also necessary to include structural forces that restrict or resist migration in and between regions of origin and destination and the aspirations of actors who, by staying put, respond to these same forces [8]. A significant research question is why certain people remain put in their homes for their whole lives. However, stayers, who have the willingness and the capacity to stay put, deserve less scholarly attention. Hence the importance of explaining why people do not migrate (immobile) is also one of the future challenges of migration theory [6].

The studies that focus specifically on non-migrants have detailed the categorization of stayers. In their study of residents living in a disaster-prone area in coastal Semarang, Amin et al. [15] identify three types of stayers based on their migration intention: the contented, the uncertain, and the discontented. Evandrou [25] provides a categorization of stayers, distinguished between intentional and unintended stayers. Categorization of stayers advances the goal of searching inside the category of immobility by relying on the process of staying itself.

Immobility should be considered a process with its own determinants, that is, as a complex, dynamic, diverse, and continuous phenomenon as mobility phenomenon, to significantly integrate immobility into migration study [8,13,26]. However, a burgeoning literature on immobility shows that for many non-migrants, staying represents and involves

agency. It is a calculated decision renegotiated and reiterated throughout life [8,13,24]. Collected data in Zacualpan by Mata-Codesal [7] challenge the current understandings of immobility due to taking no action. There is a significant amount of analytical power for migration research by open up the black box of immobility. Thus, immobility is a valid research subject of its own that deserves more scholarly attention [7,16].

### 2.2. Immobility in Response to Environmental Change

Compared to other research areas, the study of immobility in the field of environmental migration is relatively young and received little attention in academic literature [12,23]. Yet it is widely recognized that in every place that has experienced out-migration, there are many people who do not migrate because they cannot, or because they do not want to, or a combination of these two reasons [8]. Lack of interest in migration or unwillingness to migrate is less conceptualized in the environmental migration literature.

Instead of being explicitly examined as a decision in itself, immobility has provided the background against which migration occurs and the provision of a control group in environmental migration studies. For instance, the Foresight Report of 2011 acquaints the concept of 'trapped populations' to identify those most vulnerable to environmental changes but do not have the resource to migrate [4].

The increasing popularity of the term 'trapped population' as a connective thread in environmental migration studies suggests a lack of immobility agency [1]. However, the trapped population term covers the complexities of why populations stay put under challenging conditions and assume that their mobility and place attachment are homogeneous [12]. The trapped population term does not accurately describe immobility because the complexity of the narratives being collected calls for a more neutral term. Trapped begins with a more normative attitude that requires the desire to move, which is not always present. Immobility is not about the agency. Some want to move, some do not want to move, and there are those whose aspirations are not easily categorized between one of them.

Early currents studies give an insufficient explanation on how people made a decision not to migrate and why. However, in recent years, scholars have shifted to a more robust understanding of non-migrants in environmental migration study, but still without explaining their driving factor for staying. At the very least, scholars have started to realize that environmental degradation does not simply contribute to migration and can potentially limit opportunities for mobility. Unlike previous studies that almost exclusively applied to migration or displacement in the mobility study, [1,19] incorporate immobility as a result in the aftermath of a hazardous environmental change within the mobility paradigm.

The inquiry of why and how individuals do not move regarding expanding migration pressures because of environmental change brings immobility into the circle of migration studies and the hypothesis of seeing how relocation happens for a few and not others [12,23]. There is increasing participation in the literature on wider migration and refugee study, human geography, and anthropology in debates about climate change migration nexus. The debates will provide the grounded perspectives required to inform a sound interpretation of what it means to migrate or not migrate in the light of environmental degradation. This will help to put academic, political, and policy debates more closely in line with the everyday practices and heterogeneous interests of those affected [27].

Hannam, Sheller, and Urry [28] regarded the new mobilities paradigm as an analytical starting point for an extended research agenda on environmental immobility. Slowness, along with acceleration, blockages, stoppage, pressure, distribution, forced movement, and freedom of movement, must be considered an epistemological framework for studying the environmental change-immobility nexus.

Researchers conducted a study on the influencing factors of disaster immobility by building up a framework for seeing environmentally induced migration or its absence. Their framework describes the migration decision as a result of macro considerations (e.g., environmental conditions at the place of origin), micro considerations (e.g., the social status

and relationships of the potential migrant and the particular vulnerability of a migrant to environmental stressors), and a series of prevailing conditions (for example legal, logistical, or financial barriers to migration) [29].

Foresight's reports and subsequent studies established financial obstacles to migration for people with natural resource-based livelihoods. Depletion of assets will hinder normal migration responses to changes in the environment [4,30,31]. Broadly environmental degradation, specifically climate change, adds an essential aspect by worsening poverty and vulnerability. Therefore, it is essential to examine the economic aspect as pre-existing vulnerability factors in migration or immobility and feedback into the causality web [7]. Many scholars also reference various forms of capital (human, social) needed to move or deed as a hurdle to migration whether in or out of environmental contexts. Although capital constraints undoubtedly hold people in situ or keep up with their post-initial movements, the focus on economic factors must be sufficiently combined with other factors of immobility [32]. Some researchers admit the significance of social networks and the absence of emotional ties in preventing migration outside the community of origin, while others are more likely to examine the preference to stay and voluntary immobility instead of labeling 'trapped populations' [12].

The fact that immobility is never mono-causal is reiterated. In order to shape patterns and outcomes of immobility, factors do not act alone. Instead, they are related to social, economic, political, demographic, and environmental factors. In this manner, multi-causality is broadly acknowledged in clarifying the aspirations and capabilities of migration to immobility [33].

Nevertheless, the exposure of gaping disparities is one thing that links immobility in environmental contexts. People who cannot avoid environmental harm belong amongst the most susceptible members of societies, with high poverty levels, and low levels of human and social capital. However, many of the most vulnerable individuals in the world cannot avoid the sudden and direct physical impacts of disasters and stay put in a disaster-prone area. Therefore, this paper considers disaster immobility as the research object and makes an effort to develop a theoretical model suited for disaster immobility drivers through the use of the grounded theory.

*2.3. Research Review*

The importance of explaining population immobility has drawn the interest of scholarly circles. The academic community has carried out corresponding studies on population immobility. However, the studies that have been conducted are still very limited. Moreover, these studies rarely use populations living in disaster-prone areas as empirical research samples to explore the characteristics and factors driving population immobility. There are critical contrasts in conduct decisions and acknowledgment path of immobility. Whereas in different regions, the immobility of disasters that occurs is different. According to the characteristic of disaster, the current study does not systematically address the defining factors of immobility behavior of population living in disaster-prone areas. In the climate change context, the population living in coastal areas is in a critical period of stress because of disasters. How to build a systematic model to explain the population immobility in a disaster-prone area? How can the pertinence and effectiveness of policy support be refined? Ensuring the policy supports sustainability and plays a significant role in explaining population immobility in disaster-prone areas is of great importance. Special attention is given to the following points:

1. Mobility research related to the development of a migration system has provided reasonably mature results. Some researchers have laid the theoretical foundation for immobility research in the field of migration systems. However, the study of immobility is relatively young and received little attention in academic literature.
2. The research on immobility has accomplished a good stem, but there is finite research on disaster immobility, especially the theoretical research on disaster immobility

drivers. Therefore, the study on disaster immobility drivers should incorporate various aspects such as social, economic, and environmental performance.

3.  Immobility drivers research has an increasing trend over the years. However, the relevant study is not systematic and in-depth; the empirical study is exceedingly scant. The existing research is mostly qualitative research exploring the factors affecting immobility, and there is presently no substantial evidence. Although some research has been influential in immobility, the factors and mechanisms of immobility cannot be adequately determined.

These studies lay the groundwork for research targeted at people who choose to stay in the context of environmental degradation. Based on the characteristics of climate change-induced disaster and their affluence on migration, this paper aims to construct a migration hold factor model to clarify the immobility in a disaster-prone area by the grounded theory method with in-depth interviews. This study contributes a conceptual basis and references the phenomenon of immobility that occurs in disaster-prone areas.

## 3. Material and Methods

### 3.1. The Study Area

The study was conducted in Kampong Tambak Lorok, a fishing neighborhood in the coastal part of Semarang City (Figure 1). Tambak Lorok is located in a strategic location from an economic perspective. The study area is located half a kilometer east of Tanjungmas Industrial Estate and Semarang Seaport. Additionally, Tambak Lorok has adequate amenities because it is located close to the Semarang downtown. The kampong's locations are not far from public transport facilities, namely the Semarang Tawang Train Station and the Terboyo Bus Terminal, connecting nodes between cities on Java. This kampong is also located near the Kobong Market, the center of fish trading in Semarang.

Semarang is the capital city of Central Java province. This city has the lower and upper parts. A coastal region, a transitional area between land and sea processes and activities, such as marine processes, fluvial processes, and intensive human activities, is the lower part of this city. This lower part deals with three kinds of floods: local floods, river floods, and tidal flooding. The first two are mainly caused by rainfall and inadequate drainage system capacity. These first two floods are getting worst in the rainy season, which is a rainfall of almost 250 millimeters every month [34]. At the same time, tidal flooding occurs every day in some parts of the Semarang coastal area. Tidal floods are related to sea-level rise due to climate change and exacerbate with land subsidence caused by groundwater extraction [34–36]. Land subsidence in the coastal region of Semarang is around nine centimeters per year [35]. The entirety of northern Semarang is now below sea level, which will expand as the impact of tidal floods increases in the future. The rise in sea level contributes to even more significant tidal flooding, particularly in combination with land subsidence [37]. Presently, every 5 to 7 years, the residents have to raise their floor house level by more than half a meter to avoid tidal inundation [38,39].

Inhabited by 1468 households with a total population of 9715 people, Kampong Tambak Lorok suffers from a high disaster vulnerability. Some parts of the neighborhood were flooded every day due to land subsidence and tidal inundation. Moreover, in the future, the area will get worse because of the continuing land subsidence of 9 centimeters every year. Water will inundate the neighborhood, and the activities of residents will be increasingly disrupted.

In many cases, environmental factors play a role in strengthening migration decisions, especially in the most disaster-prone populations [40]. The recurrent and increasingly severe tidal inundation is the push factor for Tambak Lorok residents to migrate. However, the residents continue to stay (immobile) in this disaster-prone area, which makes it relevant to be the subject of this research.

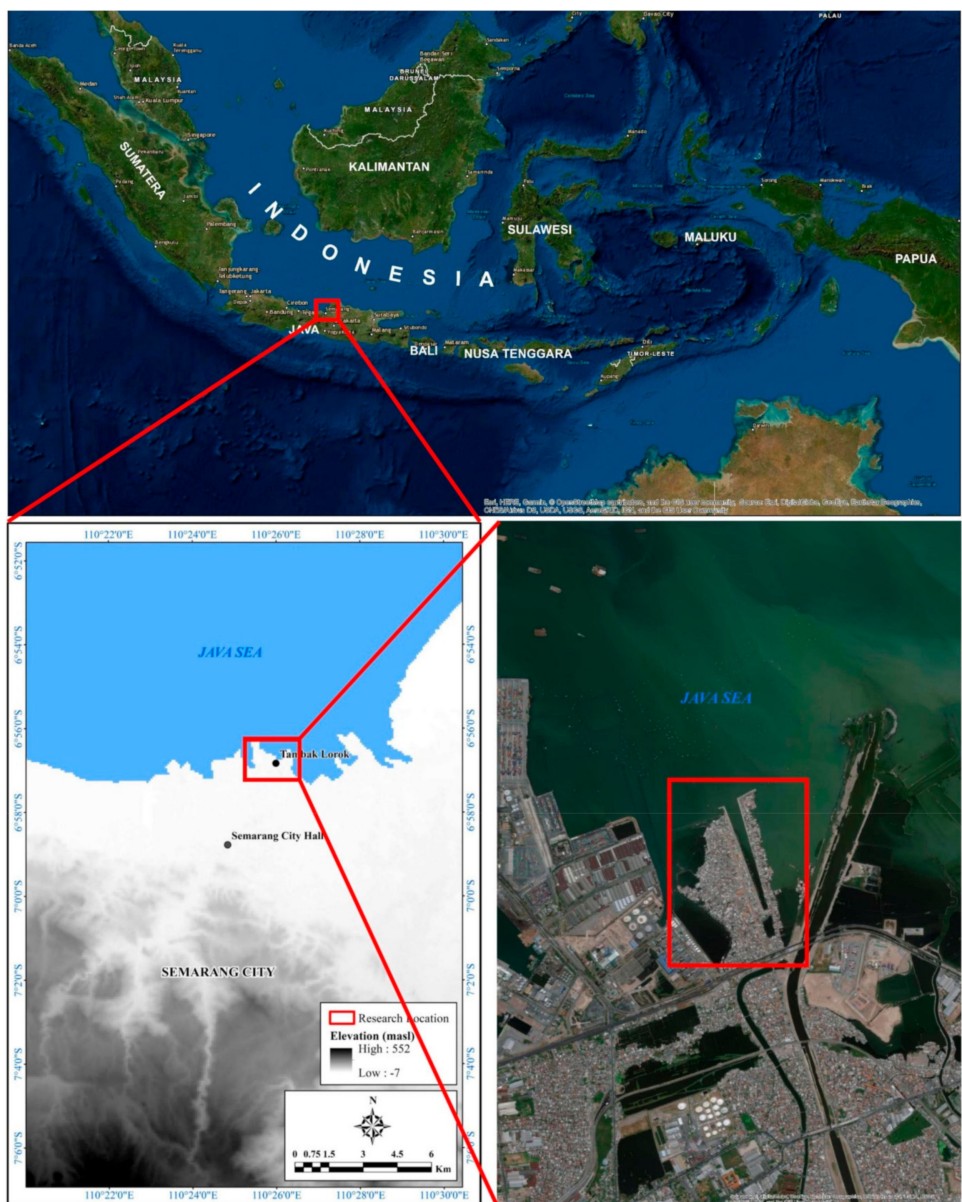

**Figure 1.** The study location.

### 3.2. Research Method

Through systematically collecting and analyzing data, the grounded theory method investigates and constructs the theory behind phenomena [41,42]. This study attends to exploring the factors holding mobility in a disaster-prone area. In this field, there is currently a shortage of mature theories. Grounded theory has essential benefits in constructing theory and is regarded as the most scientific method in qualitative research [43]. Hence, it is reasonable to employ the grounded theory to study the mobility hold factors in a disaster-prone area.

Figure 2 presents the grounded theory research procedure for this study. This research applied grounded theory to clear up and examine in-depth interview materials based on posing questions and systematic literature analysis. Open coding, axial coding, and selective coding were conducted consecutively by in-depth interview materials. Finally, the theoretical model of migration hold factors in disaster-prone areas was constructed following the theoretical saturation test passage.

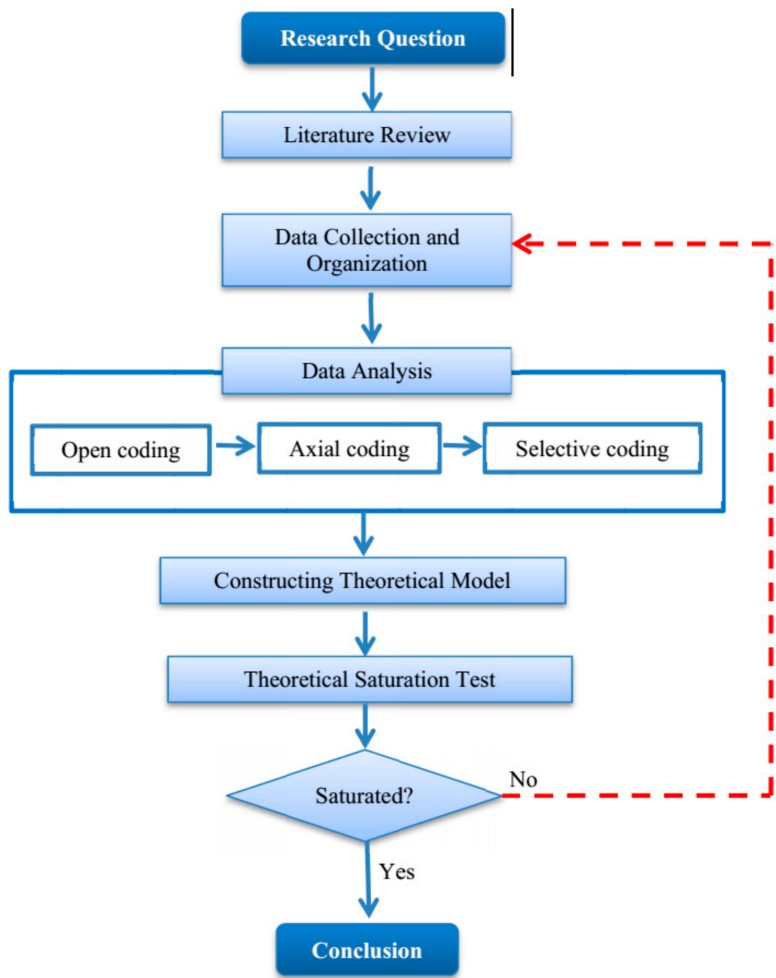

**Figure 2.** The procedure of the study.

### 3.3. Data Sources

The study subjects were carefully chosen to develop a comprehensive description of the qualitative data. Grounded theory research should be based on the theoretical sampling principle. If the case samples exhibit the "talking pig" characteristic, it is a representative sample that meets the study's requirements [44]. This study organized theoretical sampling conferring to the following two criteria, considering the sample's typicality. First, subjects lived in the study area for more than 30 years and never migrated to other areas. So, it is assumed subjects have in-depth knowledge and experience about the environmental challenges and threats in the study area. Second, the subjects' place of birth was diversified, covering natives and immigrants from outside the study area. Third, the types of subjects' livelihoods were heterogeneous, comprised of marine-related and non-marine related livelihoods. Therefore, the subjects selected in this research were typical. In-depth interviews were conducted with 24 subjects in Kampong Tambak Lorok, Coast of Semarang, Indonesia, from 7 January to 28 February 2019. With a total length of 2102 min, 24 interview recording materials were formed. In total, 75,462 words of recorded text material were obtained after being converted into text. There were 24 data samples processed until all the recorded materials were converted into text materials. Table 1 shows the study participants' demographic characteristics in terms of gender, marital status, age, place of birth, length of stay, level of education, and occupation.

**Table 1.** Demographic characteristics of the study participants.

| Characteristic | Number of Informants | Percentage |
|---|:---:|:---:|
| Gender | | |
| Female | 2 | 8.33% |
| Male | 22 | 91.67% |
| Marital Status | | |
| Married | 23 | 95.83% |
| Widow | 1 | 4.17% |
| Age | | |
| 31–40 years old | 8 | 33.34% |
| 41–50 years old | 10 | 41.66% |
| 51–60 years old | 4 | 16.66% |
| 61–70 years old | 1 | 4.17% |
| More than 71 years old | 1 | 4.17% |
| Place of birth | | |
| Inside Kampong (natives) | 12 | 50.00% |
| Outside Kampong (immigrants) | 12 | 50.00% |
| Length of stay | | |
| 31–40 years | 8 | 33.34% |
| 41–50 years | 12 | 50.00% |
| 51–60 years | 2 | 8.33% |
| 61–70 years | 2 | 8.33% |
| Level of education | | |
| Primary school | 16 | 66.67% |
| Junior high school | 0 | 0.00% |
| Senior high school | 6 | 25.00% |
| Bachelor degree | 2 | 8.33% |
| Occupation | | |
| Marine-related job | 12 | 50.00% |
| Non-marine-related job | 12 | 50.00% |

The following results are shown in Table 1: (1) In the survey participants, the percentage of men and women was 91.67% and 8.33%, respectively, suggesting that the male participant was higher than the female participant. (2) Most of the participants were married (95.83%) and only a small proportion were widows (4.17%); (3) the highest age distribution in the study subjects was 41–50 years old (41.66%), 31–40 years old (33.34%), 51–60 years old (16.66%), followed by 61–70 years old, and more than 71 years old (8.33% each), indicating that the middle-aged were main participants; (4) the place of birth is equal between inside and outside Kampong Tambak Lorok (50.00% each), indicating that the participant's place of birth inside Kampong Tambak Lorok as many as outside Kampong Tambak Lorok; (5) the highest length of stay distribution in the study subjects was 41–50 years (50.00%), followed by 31–40 years (33.34%), 51–60 years (8.33%), and 61–70 years (8.33%), indicating that most of the study subjects were living in the study area for a very long time; (6) the distribution of primary school degrees was the highest in the study subjects (66.67%), followed by senior high school degree (25.00%), and bachelor degree (8.33%), indicating that most of the study subjects have a primary school education; (7) the occupation distribution is equal between marine-related jobs and nonmarine-related jobs (50.00% each), indicating that the participant's occupation in the marine-related job was as many as a non-marine-related job.

## 4. Results Analysis

Following the research method and review procedures, the in-depth interview data were transcribed verbatim. Starting with open coding, axial coding, and selective coding, an intensive coding process eventually completed the data analysis process. The theoretical

saturation was tested after acquiring the conceptual model, and the conclusive theoretical model was developed and elucidated.

*4.1. Open Coding*

As part of the analysis, open coding deals specifically with labeling and categorizing phenomena through careful examination of the data. By means of the open coding process, the concepts are first obtained from a sample and grouped into categories. The original statement in a large sample was labeled "c". After that, the concept "cc" was initially obtained through the label's classification and abstraction. Lastly, the category "C" was obtained by the classification and abstraction of concepts. The samples were transformed into the same concepts and categories as their original material using conceptualization and categorization that will further refine and identify the relationships among categories. In total, 18 concepts and eight categories were acquired just after open coding of 24 cases. Table 2 shows an example of open coding.

**Table 2.** Open coding example (part).

| Primitive Statement | Initial Concept | Initial Category |
|---|---|---|
| c1. I know all the corners of this kampong precisely; c2. There are lots of great hangouts place here; c4. I know where the best food stalls are here | cc1. Familiarity with local landscape | C1. Familiarity with the place |
| c5. It is sweltering during the day, but the nights are more relaxed and more comfortable here; c6. In May, there must have been high tide. Then June is the peak of the high tide. | cc2. Neighborhood awareness | |
| . . . | | |
| c7. I was born here, and I want to die here; c8. It would be hard to leave my homeland | cc3. Place of birth | C2. Emotional attachment to the place |
| c10. When I was a kid, the beach was wide and clean. We played soccer every afternoon on the beach; c12. My childhood moments are a joy to me, and I can't forget my playground as a child. | cc4. Childhood memories | |
| c15. I am already sticky with this place. Since I was born until now, I am 58 years old I have never moved to another place; c.16 Since moving to this place 40 years ago, I never moved to another place. Therefore, I am very attached to this place. | cc5. Long-term attachment with the place | |
| . . . | | |
| c20. My child is a toddler. If I left, who will look after them; c22. My child's school is nearby. If I moved, it would be troublesome. | cc6. Parenthood | C3. Obligation to family member |
| c25. My mother is elderly, and I have to take care of her. | cc7. Ties to parents | |
| . . . | | |
| c29. My siblings are eight, and all of them live here; c31. I don't have relatives in other areas. My brothers are all here. | cc8. Ties to siblings | C4. Extended family support |
| c33. My extended family often helps me. If I need something, for example, money to pay for my child's schooling, they will gladly borrow it. | cc9. Extended family relationship | |
| c37. I am a widow. I live with my oldest child. He was the one who took care of me. | cc10. Widowhood | |
| . . . | | |

**Table 2.** *Cont.*

| Primitive Statement | Initial Concept | Initial Category |
|---|---|---|
| c40. We often hold community service to clean sewers and others; c41. After the tidal inundation receded, we usually helped each other clean the house from the trash and mud carried by the flood. | cc11. Mutual cooperation | C5. Social support |
| c46. We are always united because we often gather together in recitation groups. There are many recitation groups, women recitation groups, pilgrimage groups, and others; c47. I can't go or stay at another place for too long because I have to attend a social gathering held once a week | cc12. Social group member | |
| . . . | | |
| c52. I used to gab as much as I could until midnight with my neighbors. In other places, I may not be able to do it; c55. Already have a good relationship with neighbors, it's hard if you want to separate. It is difficult to adapt to new neighbors, isn't it. | cc13. Neighbor ties | C6. Cozy social interaction |
| c58. After returning from fishing, we always gather on the kampong substation while relaxing and chatting; c60. I am worried that if I moved to another area, the habits were different from here. | cc14. Familiarity with local habits | |
| c61. I have lots of friends here. We have been through distress and pleasure together. It was hard to leave them; c62. Even now, when I return to my hometown, I don't even feel at home there. I want to come back here quickly because friends are all here; c63. My childhood playmates are now my neighbors, so we hang out very closely | cc15. Attachment to friend | |
| . . . | | |
| c65. How could we not settle down here? There are so many sources of income around this area. For example, mothers can work sorting out green shells or become fish traders in the fish market. The younger people can work in the seaport area as factory workers or port workers, and some are fishing tour guides. Some supply the needs of ships that land at the seaport, such as providing food and drinking water; c68. Most of us only graduated from primary school. We can't work in an office. Since we live close to the sea, where else can we find a job if not fishing? Because being a fisherman does not require a school diploma. | cc16. Knowledge of the local labor market | C7. Sufficient employment opportunity |
| c73. It is easy to find a job. If you want to work in the sea, the sea is close. If you don't want to work in the sea, it's also close to the seaport; c74. There is rarely unemployment in this neighborhood because finding a job is very easy; c75. What I love about this area is how easy it is to find a job. I can easily work at sea as a fisherman or work in a seaport. | cc17. Easy to find a job | |
| . . . | | |

**Table 2.** *Cont.*

| Primitive Statement | Initial Concept | Initial Category |
|---|---|---|
| c78. My workplace is only a 10-min walk away. I sometimes walk to work, so it saves transportation costs; c79. I haven't moved because it is close to where I work: the Semarang Seaport; c82. I have tried various jobs. It turns out that the best is being a fisherman, so I am still a fisherman now. Therefore, I will still stay here close to the sea anyway; c83. Being a fisherman here is best because the place to catch fish is very close. It only takes 5 min by boat | cc18. Close to workplace | C8. Closeness to workplace |
| . . . | | |

The initial concept is continually unified and eliminated. In total, 18 initial concepts are grouped into eight initial categories: familiarity with the place, emotional attachment to the place, obligation to family members, extended family support, social support, cozy social interaction, sufficient employment opportunity, and closeness to the workplace.

*4.2. Axial Coding*

Axial coding is the means of connecting codes using a combination of inductive and deductive reasoning. By examining the inherent relationship between categories, an axial coding process categorizes and abstracts the initial categories into the main categories. Accordant with axial coding steps, this paper arranges 18 initial categories into eight main categories: familiarity with the place, emotional attachment to the place, obligation to family members, extended family support, social support, cozy social interaction, sufficient employment opportunity, and closeness to the workplace. Table 3 shows the main categories formed by axial coding.

**Table 3.** Main categories formed by axial coding.

| Initial Category | Main Category |
|---|---|
| Familiarity with the local landscape; Neighborhood awareness | Familiarity with the place |
| Place of birth; Childhood memories; Long-term engagement with the place | Emotional attachment to the place |
| Parenthood; Ties to parents; Ties to siblings | Obligation to family member |
| Extended family relationship; Widowhood | Extended family support |
| Mutual cooperation; Social group member | Social support |
| Neighbor ties; Attachment to friends | Cozy social interaction |
| Knowledge of the local labor market; Easy to find a job | Sufficient employment opportunity |
| Close to workplace | Closeness to the workplace |

Table 3 indicates that the main categories of familiarity with place are comprised of two initial categories, i.e., familiarity with the local landscape and neighborhood awareness. The main categories of emotional attachment to the place consist of three initial categories, i.e., the place of birth, childhood memories, and long-term engagement with the place. The main categories of obligation to family members consist of three initial categories, i.e.,

parenthood, ties to parents, and ties to siblings. The main categories of extended family support consist of two initial categories, i.e., extended family relationship and widowhood. The main categories of social support comprise two initial categories: mutual cooperation and social group member. The main categories of cozy social interaction consist of two initial categories, i.e., neighbor ties and attachment to friends. The main categories of sufficient employment opportunity consist of two initial categories, i.e., knowledge of the local labor market and easy to find a job. The main categories of close to workplace consist of one initial category, i.e., closeness to the workplace.

### 4.3. Selective Coding

By exploring the inherent relationship between the main categories, selective coding systematically selects categories to find core categories. This paper depicts each sequence's relationship using selective coding steps, focusing on the storyline of 'The mobility hold factor in a disaster-prone area'. Table 4 shows the main categories formed by selective coding.

**Table 4.** Core categories formed by selective coding.

| Main Category | Core Category |
|---|---|
| Familiarity with the place | Place attachment |
| Emotional attachment to the place | |
| Obligation to family member | Family ties |
| Extended family support | |
| Social support | Social ties |
| Cozy social interaction | |
| Sufficient employment opportunity | Occupational ties |
| Closeness to workplace | |

Table 4 shows that the core category of place attachment includes two main categories: familiarity with the place and emotional attachment to the place. The core category of family ties includes the obligation of family members and extended family support. The core category of social ties includes two main categories: social support and cozy social interaction. The core category of occupational ties includes two main categories: sufficient employment opportunity and closeness to workplace. The migration hold factor in a disaster-prone area is formulated by conforming to the logical relationship among the four core categories of place attachment, family ties, social ties, and occupational ties (Figure 3).

### 4.4. Saturation Test

When data saturation occurs and a sufficient theory emerges from the data, grounded theory research is concluded. Data saturation occurs as data processing no longer contributes to elaborate the phenomena being studied [45]. Following the steps of grounded theory research, the theoretical saturation test logic is to repeat the coding of the samples treated. The saturation test is passed when the data and data extracted by the sample are saturated and adequate theory is obtained [46]. The theoretical saturation test of the following 24 case materials was conducted with Nvivo software to evaluate whether the theoretical model that has been established above has reached theoretical saturation. The saturation test results indicate that the main categories, initial categories, and relationship depictions extracted are clear and robust. There is no new category and relations after the theoretical saturation test other than place attachment, family ties, social ties, and occupational ties. Of the four main categories, there is no new initial category. It can then be established that the holding factors model of immobility in a disastrous area, in theory, has reached saturation.

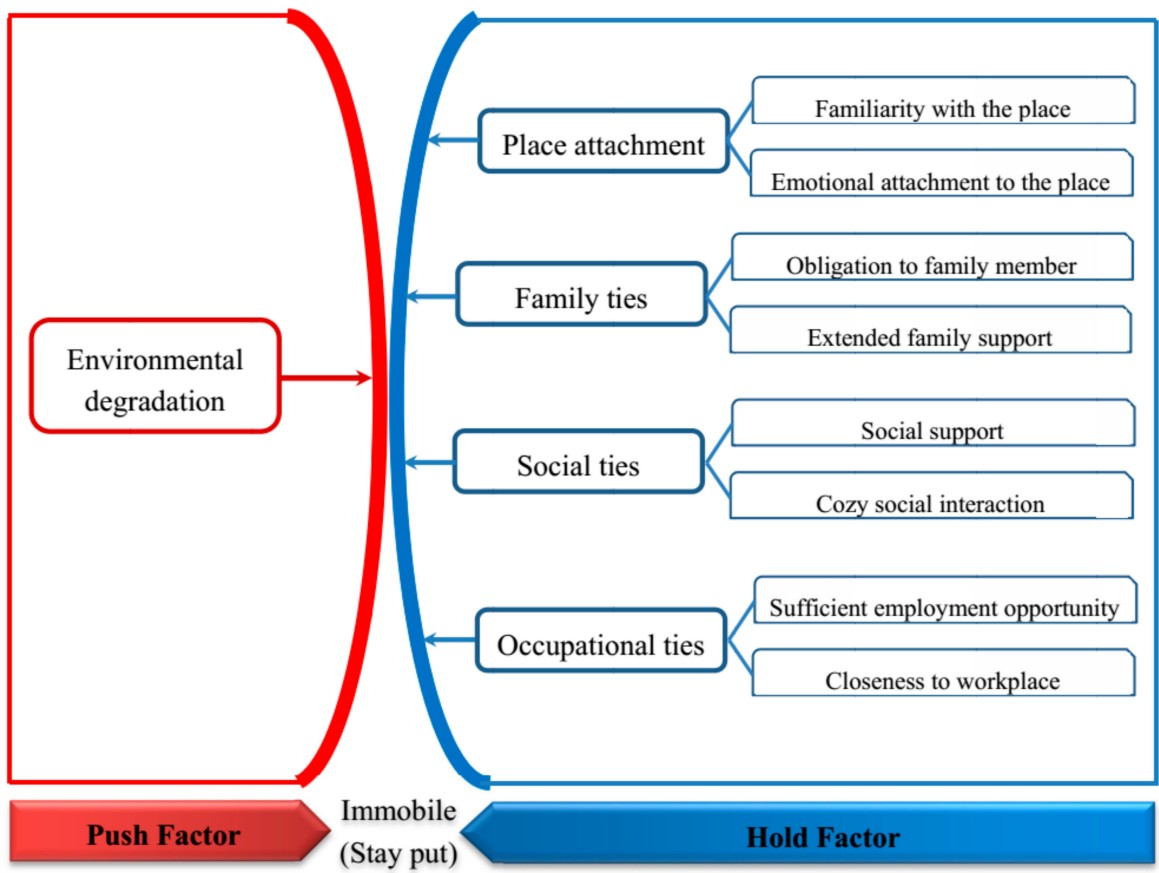

**Figure 3.** The migration hold factor in a disaster-prone area.

### 4.5. The Migration Hold Factor Model

The coding process of three levels (open coding, axial coding, and selective coding) and the theoretical saturation test have concluded that the migration hold factor model is a saturated theoretical model. Following the model (Figure 3), it is possible to obtain the following two basic propositions:

**Preposition 1.** *Population in disaster-prone areas choose not to move (immobile) because they are held by place attachment, family ties, social ties, and occupational ties.*

This proposition relates to the factors associated with the area of origin that affect the migration decision making. Based on the grounded theory of in-depth interview materials, this research explores the factors associated with the decision to stay put in a disaster-affected area. Environmental degradation plays a role in strengthening migration decisions. However, the population continues to stay in a disaster-prone area because they are held by four holding factors: place attachment, family ties, social ties, and occupational ties.

Place attachment has the potential to explain why some people choose to stay in a disaster-prone area. This finding corresponds with traditional ideas of the linkages between migration and place attachment in which positive place attachment decreases the probability of migration. It seems that strong place attachments will result in long-term immobility, that is, a choice to remain in the homeland despite potential opportunities elsewhere. Length of stay in a neighborhood had even the most significant influence on place attachment. Long-term residents are more likely to become accustomed to the neighborhood's conditions and learn how to use its resources. Individuals who stay in a place over their lifetime have very little interest in moving away from the homeland. These individuals were born, raised, and likely want to die and be buried in the homeland.

Family ties in this study include the obligation to family members and extended family support. Family ties appear to become important at certain junctures in the life course. Family members who live nearby can also be the help in emergency situations. Individuals whose siblings and parents live nearby have an obligation to take care of them. They are indeed less likely to migrate than others. Therefore, ties to family are influential agencies impeding people from migrating.

Social ties are associated with social capital and occur in a local social environment in the form of social support and cozy social interaction. The number of close friends who live nearby, having someone to turn to in an emergency, or interaction with neighbors are all examples of social ties that become social capital. Social ties may provide subtle support and are therefore widely perceived positively since they help the individual maintain community bonds. Social ties can be more powerful than physical attachments, emphasizing the role of the social environment in forming an attachment to the place of living. Such social ties have been found in this study to deter migration because the loss of social ties is a potential cost of mobility.

Occupational ties include sufficient employment opportunity and closeness to workplace. Working near to home is likely to have a positive link to the local labor market. For instance, a person who lived in areas with access to better employment opportunities and more diversity in employment should not experience locational trade-offs. As a result, a person's attachment to the local labor market or a specific job can make them hesitant to migrate.

**Preposition 2.** *Migration hold factors induce immobility by resisting the impulses of migration push factor.*

This proposition relates to the conceptual framework of migration hold factor in a disaster-prone area. To explain the immobility behavior, the hold factor model has the potential to enhance the push-pull framework presented by [14]. Push factors are negative values in a place that encourage people to move out of the place, so the result is 'out migration'. On the other hand, pull factors are positive values in a place that attract people to move into the place, so the result is 'in migration'. While the hold factors resulting in different output. Whereas the hold factors make people who live in a place keep staying in that place, the result is to stay put (immobile). In short, push factors generate out-migration, pull factors generate in-migration, while the hold factors encourage immobility.

For the individuals who choose to stay in a place, the push-pull framework is not suitable because the individual is not too affected by the push factors in the place, so that they are not motivated to migrate out of place. Meanwhile, these individuals are not affected by the pull factors in other places, so they are not interested in migrating to another place. Therefore, for the individuals who choose to stay in a place, they need another conceptual framework to explain their immobility behavior.

Migration push factors in the form of environmental degradation can trigger migration behavior. However, with the presence of a hold factor, migration may not occur because the hold factor resists the push factor's impulse. In every area, there are holding factors that act to bond people within the area. People choose to stay (immobile) in the context of increasing migration pressures because the migration holding factor holds them.

## 5. Conclusions, Implications, and Limitations

### 5.1. Conclusions

Two main findings can be put forward as conclusions from this research, namely:

(1) Populations who stay put in disaster-prone areas are held by four holding factors: place attachment, family ties, social ties, and occupational ties.

(2) Migration hold factors generate immobility by resisting the forces of migration push factor. Integrating hold factors as drivers of disaster immobility improves the environmental immobility theory, migration theory, and research on environmental migration.

Furthermore, there are theoretical contributions and practical value to this article. (1) Theoretical contribution: First, it enhances the environmental immobility theory, migration theory, and the field of environmental migration research. Second, the migration hold factor theoretical model constructed by this study provides a reference for the following research. Third, this research extends the grounded theory's scope. (2) Theoretical contributions: First, this paper provides a reference for the government to develop sustainable policies for an immobile population who reluctant to move from disaster-prone areas. Second, this article offers a reference for disaster management strategy makers to strengthen the immobile population's resilience to environmental changes.

## 5.2. Policy Implications

When an area has immense potential for disaster and endangers its residents, the government tends to choose a re-location policy. However, another aspect that needs to be considered is the resistance or defense efforts of the community. It is expected that people who occupy an area within a certain period will defend the area. Rejection to the re-location policy will be even more substantial along with the stronger place attachments, family ties, social ties, and occupational ties. Therefore, policies related to residents living in disaster-prone areas should begin to shift from a relocation policy option to a re-building policy.

Population immobility can occur in any country, from those with strict government policies to those with no policy at all. Future climate change's potential consequences would almost certainly exacerbate an already precarious situation in all these circumstances. As disaster hits the same location frequently, observers from afar can be left wondering why residents in high-risk areas continue to stay put. Policymakers would need to find ways to reduce the vulnerabilities caused by climate change hazards, both by in situ climate adaptation plans and policies that consider the need to stay safely, as a significant number of people are and will probably become immobile.

Moving or relocating individuals, or even the entire kampong, away from danger can alter landscapes, reducing disaster risk. However, residents may refuse an offer to relocate even though they can be financially supported. As a result, there are "holdouts" who ignore to accept even the most generous financial incentives. People commonly want to stay put for reasons that money cannot buy. The migration hold factor framework provides a better understanding of immobility behavior so that it becomes a valuable reference for policy-making related to residents living in disaster-prone areas. Therefore, the awareness of migration hold factors can be used to design effective re-location programs for residents living in disaster-prone areas.

## 5.3. Research Limitations and Prospects

Despite the fact that the grounded theory method was used to create the migration hold factors model of population immobility in disaster-prone areas, this study has certain limitations. The theoretical model established in this paper only reflects the immobility of populations affected by climate change-related disasters; it has not been studied on populations affected by other types of disasters in Indonesia or other countries. Moreover, the theoretical model developed from qualitative research has yet to be empirically tested. As a consideration, the limitations of this study can be seen in two ways: First, since this analysis employs grounded theory as a qualitative research method, the model must be confirmed using quantitative methods. Second, this paper's data are from Indonesia, and the theoretical framework developed could be limited to a certain country. The next step is to gather data from all countries in order to improve and validate the model.

**Author Contributions:** C.A.: data curation, software, writing original draft preparation. S.S.: supervision, conceptualization, methodology. R.R.: supervision, writing—reviewing and editing. All authors have read and agreed to the published version of the manuscript.

**Funding:** The Doctoral Research Grant from Universitas Muhammadiyah Surakarta, grant number 740/II/2019, funded this research.

**Data Availability Statement:** Not applicable.

**Acknowledgments:** The authors wish to express their dedication to the experts who assisted them with this study. We would also like to express our gratitude to all of the reviewers who provided feedback for improving this paper.

**Conflicts of Interest:** The authors affirm no conflict of interest whatsoever. The funders did not have a role to play in the design of the study; in the collection, analysis, or presentation of the data, in the writing of the manuscript, or in the decision to publish the report.

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
