# Peer review of "Exploring Migration Hold Factors in Climate Change Hazard-Prone Area Using Grounded Theory Study: Evidence from Coastal Semarang, Indonesia"

_sustainability, doi:10.3390/su13084335_

Round 1

Reviewer 1 Report

Thank you for the opportunity of reading and reviewing your interesting article. It addresses a topic which is not exhaustively covered in the existing literature and which is fully within the scope of the journal. The research gap is clearly identified and addressed. The reserach is adequately conducted and the results are well presented and discussed.  As a minor suggestion, I think it would be beneficial to present limitations of the study, both practical and theoretical implications and future directors for study. Good luck!

Author Response

Comments: As a minor suggestion, I think it would be beneficial to present limitations of the study, both practical and theoretical implications and future directors for study.

Response: Thanks for the valuable comment. As suggested by the reviewer, we have added the limitations of the study, both practical benefits and theoretical offering, and the prospects for future study. Please refers to the revision version of the manuscript.

Reviewer 2 Report

1) It is not clear authors of the article (names and surnames);

2) The article is too long;

3) There are not aim of the research;

4) Too long theoretical part. It need to be cut;

5) There are no proposals for future in the article; 

6) Conclusions must be more specific;

7) It is not clear research methods, process of research and results.

Author Response

Point 1: It is not clear authors of the article (names and surnames);

Response: Thank you for your detail attention to our manuscript. We have added the names and surnames of authors. Please refers to the revision version of the manuscript.

Point 2: The article is too long;

ResponseThank you for the valuable comment on our manuscript. As suggested by the reviewer, we have edited the introduction section, the theoretical part, and the location study to shorten the article. As a result, the text that was previously 21 pages long is now 19 pages long. Please refers to the revision version of the manuscript.

Point 3: There are not aim of the research;

Response: Thank for your valuable comment. As suggested by the reviewer, we have mentioned the research aim in the abstract and emphasized it in the introduction section. Please refers to the revised manuscript.

Point 4: Too long theoretical part. It need to be cut;

Response: Thanks for the valuable comment. As suggested by the reviewer, we have edited the theoretical part to shorten this section. Please refers to the revision version of the manuscript.

Point 5: There are no proposals for future in the article; 

Response: Thank you for your valuable comment. As suggested by the reviewer, we have provided the prospects for future study. Please refers to the revised manuscript.

Point 6: Conclusions must be more specific;

Response: Thank you for your valuable suggestion. As suggested by the reviewer, we have made more specific conclusions. Please refers to the revised manuscript.

Point 7: It is not clear research methods, process of research and results.

Response: Thank you for your conscientious review. As suggested by the reviewer, we have edited the research methods, research process, and results to clarify more. Please refers to the revision version of the manuscript.

Reviewer 3 Report

This research paper, apparently coming from a doctoral research, deals with the construction of a theoretical model using the grounded theory method. The authors' aspiration is to promote a theoretical model to explore the "black box" of disaster immobility. Their case-study area is successfully chosen, the research questions are clear and the method to approach them are adequate ( local survey etc.).

My only recommendation to the authors is to try to carefully re-write their paper so as to enhance its readability and make to more attractive to the readers. Even if English is good, there are many inconsistencies and repetitions ( an example is the repetition of the expression "this study" or "this research" in a sole paragraph, see lines 557-570. ). Another problem is the fragmentation of the text in very small paragraphs (please see for example lines 149-175).

I think that this paper deserves publication since it is scientifically sound and has important policy implications. It concerns a critical climate change related issue. However, I would kindly ask the authors to substantially improve the presentation of their research, including their abstract that should, on my opinion, be shortened and become more concise and precise. 

Author Response

Point 1: My only recommendation to the authors is to try to carefully re-write their paper so as to enhance its readability and make to more attractive to the readers. Even if English is good, there are many inconsistencies and repetitions ( an example is the repetition of the expression "this study" or "this research" in a sole paragraph, see lines 557-570. ). Another problem is the fragmentation of the text in very small paragraphs (please see for example lines 149-175).

ResponseThank you for your detailed attention to our manuscript. As suggested by the reviewer, we have improved the paper to enhance its readability. We avoided the repetition of the expression “this study” or “this research”. We fixed up the text's fragmentation by combining the paragraph into a more coherent paragraph. Please refers to the revision version of the manuscript.

Point 2: I think that this paper deserves publication since it is scientifically sound and has important policy implications. It concerns a critical climate change related issue. However, I would kindly ask the authors to substantially improve the presentation of their research, including their abstract that should, on my opinion, be shortened and become more concise and precise.

Response: Thank you for your valuable suggestion. As suggested by the reviewer, we have shortened the abstract to become more concise. Please refers to the revised manuscript.

Round 2

Reviewer 2 Report

1)  In the line nr. 13 You wrote about  your "study". In the line nr. 613 You deleted "study" I think it is better to leave the "study" as research background.  

2) In the line 28 it is better to leave "this research";

3) in my opinion article could be shorter. I would think that some things in the “ grounded theory” explaining what is done in a study may be unnecessary. But if the authors find it significant, let it remain so;

4) there are 2 suggestions in the text of article.

Author Response

Comment 1

In the line nr. 13 You wrote about  your "study". In the line nr. 613 You deleted "study" I think it is better to leave the "study" as research background.

Response: Thanks for the valuable comment. As suggested by the reviewer, we have reused the word 'study' in research background. Please refers to the revision version of the manuscript.

Comment 2

In the line 28 it is better to leave "this research".

Response: Thank you for the valuable comment on our manuscript. As suggested by the reviewer, we keep using “this research” in line 28. Please refers to the revision version of the manuscript.

Comment 3

In my opinion article could be shorter. I would think that some things in the “grounded theory” explaining what is done in a study may be unnecessary. But if the authors find it significant, let it remain so.

Response: Thank you for your valuable comment. As suggested by the reviewer, we have shortened the article by eliminating some explanation about 'grounded theory.’ Please refers to lines 343-348 in the revised manuscript.

Comment 4

There are 2 suggestions in the text of article.

Response: Thanks for the valuable comment. As suggested by the reviewer, we have reused the word 'study' as suggested in the article's text. Please refers to the revision version of the manuscript.